# Zinc and Autophagy in Age-Related Macular Degeneration

**DOI:** 10.3390/ijms21144994

**Published:** 2020-07-15

**Authors:** Janusz Blasiak, Elzbieta Pawlowska, Jan Chojnacki, Joanna Szczepanska, Cezary Chojnacki, Kai Kaarniranta

**Affiliations:** 1Department of Molecular Genetics, Faculty of Biology and Environmental Protection, University of Lodz, 90-236 Lodz, Poland; 2Department of Orthodontics, Medical University of Lodz, 92-216 Lodz, Poland; elzbieta.pawlowska@umed.lodz.pl; 3Department of Clinical Nutrition and Gastroenterological Diagnostics, Medical University of Lodz, 90-647 Lodz, Poland; jan.chojnacki@umed.lodz.pl (J.C.); cezary.chojnacki@umed.lodz.pl (C.C.); 4Department of Pediatric Dentistry, Medical University of Lodz, 92-216 Lodz, Poland; joanna.szczepanska@umed.lodz.pl; 5Department of Ophthalmology, University of Eastern Finland and Kuopio University Hospital, 70211 Kuopio, Finland; kai.kaarniranta@kuh.fi

**Keywords:** age-related macular degeneration, AMD, zinc, autophagy, melanosomes, lipofuscin

## Abstract

Zinc supplementation is reported to slow down the progression of age-related macular degeneration (AMD), but there is no general consensus on the beneficiary effect on zinc in AMD. As zinc can stimulate autophagy that is declined in AMD, it is rational to assume that it can slow down its progression. As melanosomes are the main reservoir of zinc in the retina, zinc may decrease the number of lipofuscin granules that are substrates for autophagy. The triad zinc–autophagy–AMD could explain some controversies associated with population studies on zinc supplementation in AMD as the effect of zinc on AMD may be modulated by genetic background. This aspect was not determined in many studies regarding zinc in AMD. Zinc deficiency induces several events associated with AMD pathogenesis, including increased oxidative stress, lipid peroxidation and the resulting lipofuscinogenesis. The latter requires autophagy, which is impaired. This is a vicious cycle-like reaction that may contribute to AMD progression. Promising results with zinc deficiency and supplementation in AMD patients and animal models, as well as emerging evidence of the importance of autophagy in AMD, are the rationale for future research on the role of autophagy in the role of zinc supplementation in AMD.

## 1. Introduction

Age-related macular degeneration (AMD) is the main cause of legal blindness in the elderly in the Western World, and zinc supplementation is reported in several studies to have a beneficial effect for AMD patients. However, some studies report no effect and others, rare—even an adverse influence. It is not completely clear how zinc deficiency may influence AMD pathogenesis. It is important to address this problem as the European Food Safety Authority Panel reported that a cause and effect relationship between the dietary intake of zinc and the maintenance of normal vision had been satisfactorily established (cited in [1]).

Autophagy, an evolutionary conserved process that serves the cell by removing and recycling damaged or no longer used cellular components, declines with age and its impairment is associated with several human age-related diseases, including AMD (reviewed in [2]). However, the role of autophagy in AMD pathogenesis, similar to its general role, raises many unanswered questions (reviewed in [3]).

Zinc may modulate, in many cases stimulate, autophagy, although not all details of this modulation are certain (reviewed in [4]). On the other hand, impaired autophagy may change cellular zinc homeostasis.

In summary, we can consider mutual relationships between AMD pathogenesis and zinc deficiency, impaired autophagy and zinc deficiency, as well as AMD pathogenesis and impaired autophagy. Therefore, the ternary relationship between zinc, autophagy, and AMD is logical, but it has not been shown so far. In this review, we present and update information on mutual relationships within the triad zinc–autophagy–AMD and provide arguments that at least a part of the protective effect of zinc against AMD can be attributed to the modulation of autophagy by zinc.

## 2. Age-Related Macular Degeneration

Age-related macular degeneration (AMD) is a complex eye disease that is the main cause of legal blindness in the elderly in industrialized countries. Usually, two phases of the disease are distinguished: early and late (advanced), although this categorization fails in some cases. AMD is strongly correlated with age as the prevalence of its early form in Europe increased from 3.5% in subjects aged 55–59 years to 17.6% in individuals aged greater or equal to 85 years; for late AMD, these ratios were 0.1% and 9.8%, respectively [5]. The occurrence of AMD in the United States (US) is projected to increase to 22 million by the year 2050, while the global prevalence is anticipated to increase to 288 million by the year 2040 [5,6]. The dependence on age was strengthened by the discovery of an increased level of amyloid-beta in the aging retina and AMD retinas [7]. Due to this fact, AMD is sometimes described as “dementia of the eye” or “Alzheimer’s disease in the eye” [8].

AMD affects the macula—a small structure in the center of the retina that is responsible for sharp central and color vision. AMD causes vision loss due to non-functional or dead photoreceptors and underlying retinal pigment epithelium (RPE) cells [7] (Figure 1). The RPE contains polarized epithelial cells contacting photoreceptor outer segments (POSs), while their other part contacts Bruch’s membrane (BM) between the RPE and choriocapillaris. Phenotypically, advanced AMD can be divided into two basic forms: dry (atropic, non-exudative) and wet (exudative). In its dry form, the disease is initiated by the formation of drusen and pigment mottling associated with dysfunctions in RPE cells. Dry AMD may progress to geographic atrophy (GA) associated with the loss of RPE and photoreceptors. In wet AMD, RPE remains integral but initiates the production of angiogenic agents, including vascular endothelial growth factor (VEGF) that stimulate the formation of new blood vessels from underlying choriocapillaris. This process is termed choroidal neovascularization (CNV) and may lead to photoreceptors’ death. Dysfunction of choriocapillaris is often observed in both AMD types.

Aging is the most serious risk factors associated with AMD. Familial history and genetic variability in several loci, mostly associated with the complement system, are reported to increase AMD risk [9]. Many environmental and lifestyle influences are presented as AMD risk factors, including smoking, fat-rich diet and obesity, UV and blue light exposure, female sex, light color of the iris, associated cardiovascular disease, high blood pressure, but the causality of these factors is problematic. Oxidative stress is frequently mentioned as an AMD risk factor, but it is associated with almost all the previously mentioned factors, including aging. Moreover, the retina has the highest metabolic rate among all tissues of the human body, and it produces reactive oxygen species (ROS) even in its normal functioning, and this production increases with AMD-associated changes. It was shown that RPE obtained from AMD donors displayed increased levels of ROS and a higher susceptibility to oxidative stress than RPE of non-AMD donors [10]. Therefore, it can be concluded that oxidative stress is associated with AMD, but its role in the pathogenesis of this disease requires further research to show its source and consequences.

## 3. Autophagy in Age-Related Macular Degeneration

Autophagy is an essential cellular process in which cellular components that are damaged, dysfunctional, or no longer needed (autophagic cargo) are degraded in the lysosome and recycled. This process can be categorized into macro autophagy (hereafter termed autophagy), micro autophagy, and chaperone-mediated autophagy.

Autophagy has several clearly distinguished steps (Figure 2). In the first stage, autophagy is initiated, then nucleation of the double membrane and its elongation (phagophore formation) occur, followed by the encapsulation of the cargo and fusion with completely enclosed phagophore (autophagosome) takes place. Autophagosome then fuses with lysosome to form autolysosome, in which the cargo is degraded (reviewed in [11]). Many autophagy-related proteins (ATGs) are involved in autophagy; some of them are evolutionary conserved. Mechanistic target of rapamycin (mTOR), a nutrient sensor, is a major upstream regulator of autophagy along with AMP-activated kinase that phosphorylates Unc−51-like kinase 1 (ULK1). Microtubule-associated protein light chain 3 (LC3) and γ-aminobutyric acid receptor-associated proteins (GABARAPs) form a protein family of a key significance in autophagy. The LC3/GABARAP family proteins attach to the auto lysosomal membrane and interact with various autophagy receptors, including p62/SQSTM1 (sequestesome 1).

Autophagy is not only a stress-related process but it occurs at the basal level in normal conditions to remove byproducts of normal metabolism and is involved in many life processes. As Daniel Klionsky said, “Autophagy participates in, well, just about everything” [12].

As mentioned, oxidative stress in the retina associates AMD. Reactive oxygen and nitrogen species damage cellular biomolecules, including proteins that can be misfolded by oxidation and glutathione conjugation [13]. This kind of protein damage evokes the unfolded protein response (UPR) and can be repaired by molecular chaperones. However, if this system is not efficient, soluble proteins are ubiquitinated and targeted to degradation in the proteasome (reviewed in [14]). Furthermore, proteasomal degradation may fail or dysfunction in aging and neurodegeneration, which may lead to the accumulation of oxidized and ubiquitinated proteins (reviewed in [15]). The ubiquitination attracts autophagy receptors, such as p62/SQSTM1 and LC3, that link these cellular waste to autophagy. When autophagy fails in RPE cells, this may lead to the accumulation of lipofuscin and activation of the NLRP3 (NLR family pyrin domain-containing 3) inflammasome, eventually resulting in drusen formation and low-grade chronic inflammation in the retina, accelerating the aging process (reviewed in [16]). However, Kosmidou et al. pointed to the need for re-interpretation of published results reporting NLRP3 expression and upregulation in human and human-derived RPE cells and addressed the role that the inflammasome plays in AMD pathogenesis [17].

Non-exudative (dry) AMD seems to be of special interest from the perspective of autophagy as this form of the disease is associated with the accumulation of lipofuscin on the RPE. Lipofuscin contains lysosomal insoluble pigment granules that remain after lysosomal digestion [18]. Impaired autophagy favors lipofuscin formation, and it has been observed that rapamycin, an autophagy inducer, decreased lipofuscin accumulation in RPE cells [19]. Rapamycin also ameliorated cellular damage in the RPE induced by A2E, a lipofuscin fluorophore [20].

As autophagy contributes to the maintenance of cellular homeostasis, its high level is observed in stress conditions [21]. However, such an increase in autophagy is often observed along with cellular death, which led to the conclusion that besides pro-life functions, autophagy may act as a cellular death executioner (pro-death functions, reviewed in [22]). The mechanism of the regulation of these two pathways of autophagy is not completely clear, and it is not easy to relate them to AMD pathogenesis. As suggested by Mitter et al., autophagy might be dysregulated in two ways in AMD [19]. They observed that acute oxidative stress might increase autophagy in RPE cells, while chronic oxidative stress was associated with reduced autophagy. Therefore, when oxidative stress was induced in RPE cells, it might induce autophagy that acted in two phases—the initial increase followed by a decrease. In conclusion, Mitter et al. postulated that autophagy was important for the protection of the RPE against oxidative stress, and lipofuscin accumulation and impaired autophagy might aggravate oxidative stress and in this way play an important role in AMD pathogenesis.

## 4. Zinc in Age-Related Macular Degeneration

Zinc is an essential nutrient and a critical element in the structure and function of many protein complexes (reviewed in [23]). It is the second most abundant transition metal, after iron, in the human organism. The human body comprises about 2 g of this element, with more than 99% located within the cells [24]. Although the actual distribution of zinc within a cell is not easy to determine, about half of its amount can be found in the cytoplasm, while the remaining half is distributed between the nucleus and cytoplasmic membrane in a ratio of about 3–4:1–2 [25]. Zinc deficiency is an emerging global problem, as nearly 2 billion people are at risk of such syndromes [26]. In most physiological conditions in humans, zinc is maintained in the Zn(II) (Zn^2+^) state [27]. In humans, as in other mammals, zinc can be either bound to proteins or occur in a “free” state, which represents zinc ions bound to non-protein ligands (reviewed in [28]). Such “free” or transiently bound zinc is reported to be present in the endoplasmic reticulum, Golgi apparatus, and mitochondria. Such organelle-compartmentalization of zinc is essential for its cellular homeostasis and functionality.

Zinc is distributed in well-regulated gradients relative to the plasma membrane and intracellular compartments [29]. Similar zinc gradients are also within the cells. Zinc homeostasis is regulated by proteins that are dedicated to Zn^2+^ transport and buffering (Figure 3). First of all, two groups of zinc transporters belonging to two solute carrier families: at least ten members of the ZnT (Zn^2+^ Transporter) family and 14 members of the ZIP (Zn^2+^-regulated metal transporter, Iron-regulated metal transporter-like protein) family (22), and three distinct isoforms of metallothionein. ZIPs and ZnTs transport zinc into and out of the cytosol, respectively (reviewed in [23]).

Zinc presents chemical compatibility with several ligands present in histidine, aspartate, glutamate, and cysteine residues in many proteins. Unlike other transition metals, zinc lacks biological redox activity and facilitates bound water deprotonation by lowering water pKa, which makes it a good enzymatic cofactor [30]. Therefore, the chemical properties of zinc may be important for its cellular functions. Zinc has been identified as a cofactor of hundreds of enzymes belonging to all enzymatic classes [30]. Moreover, enzyme-bound zinc plays a catalytic role and is directly involved in catalysis and is found as a structural component in thousands of human proteins, constituting about 10% of the human proteome [31].

Zinc is also reported important in the pathophysiology of several groups of diseases, including neurodegenerative diseases and cancer.

The majority of zinc present in the human body is in skeletal muscle and bone mass—approximately 60% and 30%, respectively [24]. The eye has a relatively high content of zinc among other organs and tissues with the peak concentration in the RPE about 300 µg/g dry tissue [32]. In the retinal tissue, zinc is mainly stored in intracellular compartments in ganglion cells, the horizontal and amsacrine cells as well as retinal Müller cells [33]. In addition, zinc can be present in photoreceptor outer segments (POSs), that are degraded by RPE cells [34]. As endogenous zinc is co-released with glutamate from synaptic terminals of photoreceptors, it can possibly protect the retina from glutamate toxicity [35].

Zinc deficiency is often caused by malnutrition, especially in the elderly (reviewed in [36]). Therefore, a low zinc diet in the elderly might contribute to AMD pathogenesis. Some studies suggest that the content of zinc in human AMD RPE/choroid can be lowered by 24% [37,38]. Therefore, diet supplementation with zinc may be an efficient and inexpensive strategy for AMD prevention. In the first prospective, randomized, double-masked, placebo-controlled oral intervention study in 151 subjects with drusen or macular degeneration, Newsome et al. showed that the zinc-treated group had significantly less visual loss than the placebo group after a follow-up of 12 to 24 months [39]. After this positive input, several large clinical trials were conducted. Among them, the large randomized placebo-controlled AREDS/AREDS2 (Age-Related Eye Disease Studies) are likely the best known due to commercial AREDS supplement formulation [40,41]. These studies evaluated supplementation with relatively high doses of vitamin C, E, and beta carotene with or without zinc administrated in the form of zinc oxide at 80 mg daily and copper. Then xanthophyll carotenoids, lutein, and zeaxanthin were added with or without omega − 3 fatty acids. These studies suggested that zinc supplementation in combination with AREDS formula and other antioxidants might suppress retinal degeneration [42]. Other large population-based cohort studies, the Rotterdam Eye Study and the Blue Mountain Eye study confirmed beneficial effects of higher intake of dietary zinc on incident AMD [43,44,45].

As mentioned, AMD is a complex disease with both genetic and environmental factors involved in its pathogenesis. The most convincing evidence of the role of genetic factors comes from the identification of the major AMD susceptibility alleles, including the gene of complement factor H (CFH), age-related maculopathy susceptibility 2 (ARMS2), pleckstrin homology domain-containing A1 (PLEKHA1), hypothetical gene LOC387715, located in 10q26, C2-CFB (complement C2-complement factor B), and C3 (complement C3) and HtrA serine peptidase 1 (HTRA1) (reviewed in [46]). The effect of zinc on AMD incidence may be modulated by the individual genetic background. It was noted in the Rotterdam Study that zinc supplementation was proficient for patients bearing a high-risk allele of the CFH gene [47]. On the other hand, patients of the AREDS study received a maximal benefit from the AREDS supplement when they had the ARMS2 risk allele and not a CFH risk variant [48,49]. However, these studies were questioned due to errors in statistical analysis, and it was postulated that zinc supplementation might be beneficial in AMD patients independently of genotype [50]. This was not the only paper questioning the conclusions from AREDS and stressing the conflicting results between zinc supplementation trials and epidemiological studies [51,52,53,54]. It seems that the main objection against the general value of AREDS results is the fact that the study was conducted in the US, and it is questionable whether they can be applied to populations with different dietary habits and hence, different microbiome. One way or the other, observational studies have not brought unequivocal conclusions on the beneficiary potential of zinc supplementation in AMD. Therefore, further research is needed to determine the role of this element in AMD pathogenesis. Apart from strictly nutritional aspects, molecular mechanisms should also be considered to establish this role.

As the complement component of the innate immune system has been implicated in AMD pathogenesis, and on the other hand, zinc was reported to oligomerize and inactivate CFH, it seems reasonable to search for the mechanisms of the potential involvement of zinc in AMD pathogenesis through CFH (reviewed in [1]). Besides CFH, zinc can bind to other proteins of the complement system and, thus, affect the complement activity in several modes and, in consequence, play a role in modulation of inflammation [55,56]. Therefore, zinc may affect complement-mediated inflammation that is a major pathway in AMD pathogenesis, but the exact mechanism of such action and its consequences are not known [16]. However, as zinc can function as an anti-inflammatory agent (reviewed in [57]), it could be possible that it can exert its protective role against AMD through stopping or decreasing low-level, chronic inflammation in the retina (“para-inflammation”) typical for AMD [58].

Two features of zinc functioning in the eye may be related to AMD pathogenesis. First, zinc is present in higher concentrations in sub-retinal epithelial deposits (drusen), an established early hallmark of AMD [59,60]. Second, RPE cells in the macula may contain less zinc than their counterparts in the peripheral parts of the retina, and the level of zinc in the macula may be affected by aging [61]. On the other hand, the choroid is the region in the eye with the highest zinc concentration, but the significance of this fact is not completely known [62].

As mentioned, increased oxidative stress is frequently associated with AMD, but neither its source nor its casual mechanism is known. However, zinc properties as an antioxidant in the context of oxidative stress-associated AMD pathogenesis is worth considering. Reactive oxygen species (ROS) that are produced during oxidative stress may be major executors of the stress as they can damage biological macro molecules and cellular components. Zinc shows antioxidant properties, so it can be considered in lowering oxidative stress related to AMD. The antioxidant properties of zinc result from its several features, including the inhibition of reduced nicotinamide adenine dinucleotide phosphate (NADPH) oxidase, the involvement as a cofactor of superoxide dismutase (SOD), the competition with Fe^2+^ and Cu^2+^ ions for binding to the cell membrane and proteins, the involvement in the generation of metallothionein, a scavenger of hydroxyl radical, the upregulation of nuclear factor erythroid 2-related factor 2 (NRF2), a master regulation of antioxidant enzymes and proteins (reviewed in [57,63]). Each of these antioxidant activities can be related to specific pathways of oxidative stress-related pathogenesis of AMD (reviewed in [64]).

It was suggested that zinc is essential for the transactivation function of the PGC − 1α (peroxisome proliferator-activated receptor gamma coactivator 1-alpha)/NRF2 signaling pathway [65]. We recently showed that mice lacking expression of both these proteins developed phenotype resembling AMD [66]. As will be discussed later in more detail, NRF2 may play an important role in protecting of RPE cells against oxidative stress with the involvement of zinc and autophagy. Moreover, we showed that PGC − 1α might protect RPE cells of the aging retina against oxidative stress-induced degeneration through the regulation of senescence and mitochondrial quality control [67].

## 5. Zinc and Autophagy

Autophagy is one of the critical cellular reactions to starvation to reclaim parts of proteins to use in life-preserving functions. Starvation is usually due to lack of certain nutrients, such as carbon/glucose or nitrogen, but Kawamata et al. were the first to show that zinc deficiency can also be an autophagy trigger in yeast [68,69]. On the other hand, Shinozaki et al. recently showed that autophagy maintained zinc pools increased zinc bioavailability and retained ROS homeostasis under zinc deficiency in plants [70]. These cases are not directly associated with the main subject of this review, but they show that autophagy, which is considered as an evolutionary conserved process, may be regulated by zinc and vice versa—zinc homeostasis can be regulated by autophagy. This shows that the identification of the role of autophagy in a given process may be complex due to difficulties in determining the causal relationship between them.

Hwang et al. showed that tamoxifen, a frontline drug in hormonal therapy in breast cancer, induced vacuole formation and death of breast cancer cells MCF − 7 [71]. They also observed an increased level of LC3-II and an accumulation of GFP-LC3 in and around vacuoles in MCF − 7 cells exposed to tamoxifen, suggesting the involvement of autophagy in the observed processes. Further, the authors observed an accumulation of labile zinc ions in most acidic LC3 autophagic vacuoles. Zinc chelation blocked the increase in LC3-II level, but its addition potentiated autophagy. Therefore, zinc may be involved in the regulation of autophagy induced by an external stressor.

Lee and Koh observed an increase in zinc levels in autophagic vacuoles, including autolysosomes, which preceded lysosomal membrane permeabilization and cell death in cultured brain cells exposed to oxidative stress [72]. They concluded that the source of zinc was redox-sensitive zinc-binding proteins, including metallothioneins (MTs), which release zinc in oxidative conditions. This was confirmed by the observation that in acute oxidative stress, zinc dyshomeostasis and lysosomal membrane permeabilization are diminished in metallothionein − 3 null cells. Therefore, zinc may be released by proteins in oxidative stress and enter autolysosomes to influence autophagy.

Hung et al. treated PC12 cells derived from a pheochromocytoma of the rat adrenal medulla and cultured neurons expressing EGFP-LC3 with dopamine and Zn^2+^ [73]. They observed an increase in fluorescent puncta and LC3 lipidation in the cytosol of PC12 cells. Decreasing ATG7 level inhibited puncta formation, suggesting that treatment of PC12 cells with dopamine and Zn^2+^ results in the activation of the autophagy pathway, presumably to increase cell survival. Liuzzi and Yoo observed that zinc depletion resulted in autophagy suppression, and zinc addition caused autophagy stimulation in VL − 17A human hepatoma cells [74]. When an excess of zinc was combined with ethanol, it exerted an additive effect on the induction of autophagy. Ethanol and 3-methyladenine (3-MA), an autophagy inhibitor, induced changes in the expression of metallothionein and zinc transporters. These results support the hypothesis that zinc is essential for autophagy in basal conditions and during stress. The studies of Hung et al. and Liuzzi and Yoo also demonstrated that phosphorylation of extracellular-signal-regulated kinases (ERK1/2) is needed for zinc to regulate basal and induced autophagy [73,74].

ERK1/2 can activate autophagy by the stimulation of the Beclin 1/PI3K complex through the phosphorylation of Bcl − 2, a negative regulator of Beclin 1 [75]. Alternatively, active ERK1/2 can stimulate autophagy through stimulation of the breakage of the mTORC1 (mTOR complex 1) complex [76].

As zinc is a cofactor in many transcription factors, it is justified to assume that zinc regulates the transcription of genes essential for autophagy. This can be achieved by the activation of the metal-responsive transcription factor 1 (MTF1) that controls the transcription of two classes of genes: metal homeostasis and anti-oxidative response genes (reviewed in [77]). The latter is associated with induced autophagy, whereas the former is involved in the activation of the expression of MTs. In general, genes having binding sites for MTF1 whose products are involved in autophagy, which may be induced by zinc. For instance, the promoter of the ATG7 gene contains four sites for MTF1 located in the stretch of 2 kb upstream of the transcription start site as predicted by bioinformatics analysis, but not confirmed experimentally yet ([4], www.ncbi.nlm.nih.gov/nuccore/NC_000003.12?strand=1&report=genbank&from=11272309&to=11557665, accessed June 20, 2020).

Another link connecting zinc with autophagy is the maintenance of genome stability, which is essential for healthy aging (reviewed in [78]). Although autophagy is a cytoplasmic process, it is clear that it plays a key role in genome stability maintenance (reviewed in [79]). Autophagy can be involved in DNA maintenance directly through selective degradation of nuclear components or indirectly by modulation of an excess of ROS produced by faulty mitochondria through their degradation in mitophagy. In fact, the role of autophagy in DNA damage response and, in consequence, in the maintenance of genomic stability is an emerging issue, and the mechanisms lying behind them are still being investigated. On the other hand, zinc is an essential element in many proteins that play a central role in genome maintenance. First, DNA repair proteins (reviewed in [80]). Sharif et al. showed that this role is strongly dependent on dietary zinc [81]. Therefore, both zinc and autophagy are important for genomic stability, but their potential interaction in this respect is still to be elucidated.

The p53 protein, encoded by the *TP53* gene, plays a major role in the maintenance of genomic stability and prevents cancer transformation, but its mutated form, mutp53, occasionally avoids proteolytic degradation. Such a mutated form of p53 adopts misfolded and denaturated conformation and accumulates in malignant tumors [82,83]. Garufi et al. showed that zinc supplementation restored the proper folded conformation of p.175R > H mutp53 and its ability to bind DNA and transactivate its downstream target genes [84]. In their next work, Garuti et al. showed that a zinc compound induced degradation of p.175R > H mutp53 protein through autophagy [85]. Pharmacological or genetic inhibition of autophagy prevented Zn(II)-mediated degradation of p.175R > H mutp53H175 and restoring wtp53 DNA-binding and transcription activity. Moreover, autophagy seemed to be the main, if not the only, pathway of p.175R > H mutp53 degradation as inhibition of the proteasome did not result in any changes in the degradation outcome. Moreover, zinc restored the ability of p53 to induce the expression of its target gene DRAM (damage-regulated autophagy modulator), a crucial controller of autophagy, resulting in autophagy induction. Altogether, this work showed that zinc reactivated p.175R > H mutp53 to bind DNA and transactivate its target genes in a pathway involving p53-mediated autophagy.

Luizzi et al. underlined the special role of the zinc transporter ZnT10 in the interplay between zinc and autophagy [4]. It was reported to be located in the Golgi apparatus that may provide membranes for the formation of autophagosomes [86]. It was shown to be downregulated in human hepatoma cells after exposure to 3-MA [74]. Alzheimer’s disease (AD), a neurodegenerative disease, is reported to associate with defective autophagy [87]. It was suggested that raising cAMP (3′,5′-cyclic adenosine monophosphate) and free zinc levels in brain cells may be beneficial in normalizing lysosomal pH and autophagic flux in proteinopathic neurodegenerative disorders, including AD [88].

Lee et al. investigated the role of the dermis zinc transporter, ZIP13, in fibrosarcoma progression in dermal fibroblasts obtained from wild-type and ZIP13-KO mice [89]. They observed impairment in autophagy in ZIP13^−/−^ cells associated with low expression of LC3 at the mRNA and protein levels. As the authors observed recovery of impaired autophagy after treatment with 5-aza−2′-deoxycytidine (5-aza), which is a DNA demethylating agent, they concluded that observed changes might be underlined by epigenetic regulation, the more that they observed a decreased activity of DNA methyltransferases. Autophagy inhibitors slowed down the growth of fibrosarcoma. Therefore, epigenetic control over autophagy and zinc homeostasis can play a role in fibrosarcoma progression. Ni et al. showed that the expression of zinc transporter 4 (ZnT4) and LC3 in rat cerebral cortex increased following neonatal seizures [90].

It was observed that excess of copper or free fatty acid induced impairment of autophagic flux in human hepatocytes at the step of autophagosome-lysosome fusion, but zinc ameliorated that impairment via the maintenance of ER homeostasis [91]. It was shown that the membrane-permeable zinc chelator N,N,N,N-Tetrakis(2-pyridylmethyl)-ethylenediamine (TPEN) induced cell death, via increasing ROS production and restraining autophagy in pancreatic Panc − 1 cells [92]. Autophagy impairment was attributed to lysosomal disruption by TPEN.

Supplementation of the diet of rats with ZnCl2 shortly before bilateral ischemia with subsequent reperfusion resulted in decreased ER stress, reduced inflammation, and low expression of the autophagy-related proteins Beclin 1 and LAMP − 2 [93]. Therefore, zinc supplementation may ameliorate the negative effects associated with ischemia and reperfusion, and autophagy may be involved in the mechanism of this effect. A similar effect was demonstrated by Bian et al., who subjected rat myoblastic H9c2 cells to hypoxia and subsequent reoxygenation [94]. The treatment of the cells with ZnCl2 induced autophagy, evidenced by an increased LC3-II/LC3-I ratio and GFP-LC3 puncta. Furthermore, these authors showed that endogenous zinc was needed for the autophagy induced by starvation and rapamycin. That study also showed that zinc induced mitophagy as it upregulated several proteins related to this process. However, zinc did not induce mitophagy in cells transfected with PINK1 (PTEN-induced kinase 1) siRNA and stabilized PINK1 in mitochondria. Further experiments on ROS generation and mitochondrial membrane potential suggested that zinc prevented mitochondrial oxidative stress through mitophagy. The authors concluded that zinc induced mitophagy through PINK1 and Beclin 1 via ERK, leading to the prevention of mitochondrial ROS generation in hypoxia/reoxygenation conditions.

A high-zinc diet was reported to stimulate growth performance in pigs, but induced cell apoptosis in the intestinal epithelium [95]. It was also shown that Zn inhibited IPEC-J2 (intestinal porcine epithelial cells) cell proliferation and induced autophagy. Moreover, high concentrations of Zn evoked DNA fragmentation and upregulated LC3-II, and downregulated p62/SQSTM1. Treatment with 3-MA inhibited Zn-induced changes in IPEC-J2 cells. The author concluded that autophagy acted in concert with apoptosis to regulate zinc-induced IPEC-J2 cell death.

## 6. Zinc and Autophagy in AMD

Zinc protoporphyrin (ZPP) is a zinc compound present in trace amounts in red blood cells during heme synthesis (reviewed in [96]). In normal conditions, iron chelates with protoporphyrin, but in iron deficiency or impaired iron metabolism, zinc takes the role of iron, resulting in increased formation of ZPP. We and others have shown that compromised iron metabolism might play a role in AMD pathogenesis [97,98]. Therefore, ZPP may be an element of the involvement of zinc in AMD pathophysiology. Chloroquinone (CQ), a 4-aminoquinoline compound, is commonly used as an antimalarial and anti-inflammatory drug (reviewed in [99]). Although its mechanism of action as a pharmaceutical is not completely known, some of its biological effects may be underlined by inhibition of lysosome functions, which are critical for autophagy [100]. In particular, CQ can inhibit acidification of the lysosome, which is essential for its ability to transport and degrade the cargo via autophagy.

As mentioned, oxidative stress might be implicated in AMD pathogenesis, although not all details of this implication are clear. In general, AMD initiation and progression are associated with increased oxidative stress, so its inhibition may be important in AMD prevention and therapy. Saito et al. investigated the influence of the activation of NRF2, a crucial component of the NRF2/Keap1 (kelch like RCH associated protein 1) axis on damage in ARPE − 19 cells [101]. They used RS9, a triterpenoid, as an NRF2 activator and found that it protected ARPE − 19 cells against NaIO3-induced oxidative damage. NaIO3 is an oxidant that causes not only damage to RPE cells but it also increases the levels of abnormal unfolded proteins that can normally be a substrate for autophagy [102,103], and that this effect was inhibited by co-treatment with ZPP or CQ. Furthermore, the authors observed that RS9 augmented autophagy flux and induced transient upregulation of p62/SQSTM1. Combined treatment of CQ and RS9 inhibited the degradation of autophagosomes. RS9 and CQ showed the same actions in light-damaged zebrafish retina as those in vitro. These results primarily show the association between the acceleration of autophagy and the cytoprotective effects of NRF2 activation in RPE cells and zebrafish retina. However, they also show that zinc may influence the reaction of the RPE cells against oxidative damage with the involvement of the autophagic pathway.

Chloroquinone at high doses and, in particular, in combination with intense light induces retinopathy [104]. Three main mechanisms may lie beyond this effect [105]. First, CQ may diffuse into lysosomes and prevent degradation of the damaged or no longer used cellular material in the process of autophagy and heterophagy [100]. Second, CQ may accumulate in the RPE and choroid and bind melanin [106]. Third, CQ may induce oxidative stress as it decreases the level of glutathione and increases lipid peroxidation [64,107]. All three mechanisms can be linked with AMD pathogenesis [108]. Peters et al. observed that CQ-treated rats showed enlargement of the space between the RPE and BM, with the accumulation of residual material from phagosomes [105].

Clioquinol (ClioQ) is a lipophilic agent that may form stable complexes with zinc and copper (II) and is applied as an antibiotic to treat diarrhea and skin infection [109]. Clioquinol, in contrary to its parent compound, CQ, can activate autophagy acting as a zinc ionophore [110]. It was observed that CQ induced vacuole formation and cell death in ARPE − 19 cells, but co-treatment with ClioQ decreased toxicity induced by CQ in a zinc-dependent manner [111]. Increases in lysosome enlargement and inhibition of autophagic flux by CQ were also distinctly reduced by ClioQ treatment. These results indicate that the lysosomal level of zinc may modulate autophagy in RPE cells supporting the potential of zinc in AMD pathogenesis.

Several studies suggest that AMD is more prevalent in white than in black populations [112]. Melanosomes in RPE undergo age-related changes, resulting in their reduced amount with a concomitant increase in the amount of lipofuscin and melanolipofuscin granules [113,114]. Melanin, an essential part of melanosomes, is an important endogenous protector against retina degeneration as it screens light from sensitive tissues [115]. Moreover, melanin sequesters metals that support oxidative reactions and scavenges free radicals produced in photochemical reactions [116,117]. However, melanin is also reported to catalyze the production of free radicals and to oxidize physiological substrates in exposure to UV and visible light [118,119]. Finally, melanin and its precursors are considered as free radicals [120]. Melanosomes are especially rich in zinc [121]. Julien et al. observed that the zinc mole fraction of melanosomes of RPE of rats with zinc deficiency was smaller than controls [122]. Zinc-deficient animals also showed a greater number of lipofuscin granules and thinner Bruch’s membrane than control rats. The authors concluded that zinc deficiency in pigmented rats produced an accumulation of lipofuscin in the RPE, an event associated with AMD pathogenesis. All these relationships between zinc and lipofuscin accumulation suggest that zinc deficiency may result in the accumulation of substrates for autophagy. Therefore, low zinc requires high autophagy in RPE, but on the other hand, this low zinc does not stimulate autophagy, which altogether may result in lipofuscin accumulation and other effects typical for AMD pathogenesis.

## 7. Conclusions and Perspectives

Several studies on AMD patients and animal models, as well as cell cultures, suggest that zinc may play a positive role in AMD pathogenesis. The exact mechanism of the role of zinc deficiency plays in AMD is not known. Due to mutual relationship between zinc and AMD, zinc and autophagy as well as autophagy and AMD, we suggest that the effect of zinc in AMD can be determined by autophagy and this may be the reason that some studies do not confirm beneficiary effects of zinc in AMD, as was shown in AREDS/AREDS2, and other studies report a negative consequence of zinc supplementation for AMD patients. However, these “negative” studies are rare, and their research design is questionable, and that is why they were not considered in the context of autophagy in AMD in this manuscript [51,52,53,54].

As shown in the Rotterdam Eye Study, the effect on zinc on the vision might depend on individual genetic background [123]. In the post-genomic era, it is tempting to relate the reaction of zinc to an individual’s complete genetic profile, but it has little sense, if any. Who could elaborate the algorithm for the use of AREDS-like formulation to prevent AMD or slow down its progression and decrease consequences? Instead, a dedicated microarray could be projected with genes that are important for AMD pathogenesis and zinc metabolism. Of course, the main problem would be with the selection of AMD pathogenesis genes. In the context of the research performed so far, AMD-susceptibility genes, genes for zinc metabolism, and genes of autophagy could be included.

Altogether, the influence of zinc on AMD patients is a complex issue as many pathways can be involved, including autophagy. Moreover, autophagy in AMD is also a complicated subject with no unequivocal determination of all mechanisms lying beyond observed and expected effects. As mentioned, it seems that the transport of melanosomes and lipofuscin accumulation may be an important connection in zinc deficiency, autophagy, and AMD. Recently, Xia et al. showed that missense mutations in the UBE3D (Ubiquitin Protein Ligase E3D) gene, which was identified to be associated with wet AMD, were linked with several adverse effects in the retina, including delayed retrograde melanosome transport, increased deposition of pigment granules, and impaired autophagy in zebrafish [124]. As shown by Julien et al., the metal ion concentration of RPE melanosomes is regulated by zinc, and reduced metal-binding activity of melanosomes is associated with degenerative processes in the retina [122]. Therefore, melanosomes and their properties, especially their zinc-binding activity, may be a key element in the triad zinc–autophagy–AMD.

## Figures and Tables

**Figure 1 ijms-21-04994-f001:**
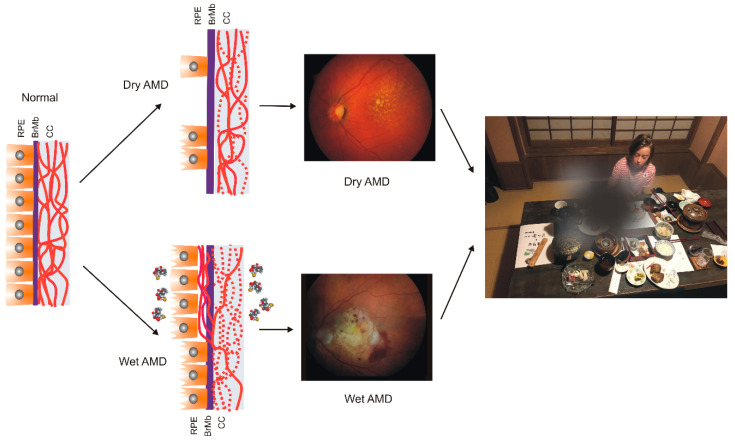
Age-related macular degeneration (AMD). The normal retina contains photoreceptors, not shown here, that interact with the retinal pigment epithelium (RPE)/Bruch’s membrane/choriocapillaris (CC) complex, which is supported by the large choroidal blood vessels (not presented). In dry AMD, some RPE cells are damaged, and some are lost when the disease progresses. Some CCs are also lost (broken lines) in dry AMD, but in the wet form of the disease, they are lost almost completely. In wet AMD, RPE cells release several factors (multicolor small objects) that support angiogenesis, including vascular endothelial growth factor, which leads to choroidal neovascularization. Fundus color images of dry and wet AMD. Light orange spots in the retina termed drusen can be observed in both dry and wet AMD, whereas wet AMD is featured by hemorrhages and edema. AMD can ruin the central vision of affected individuals.

**Figure 2 ijms-21-04994-f002:**
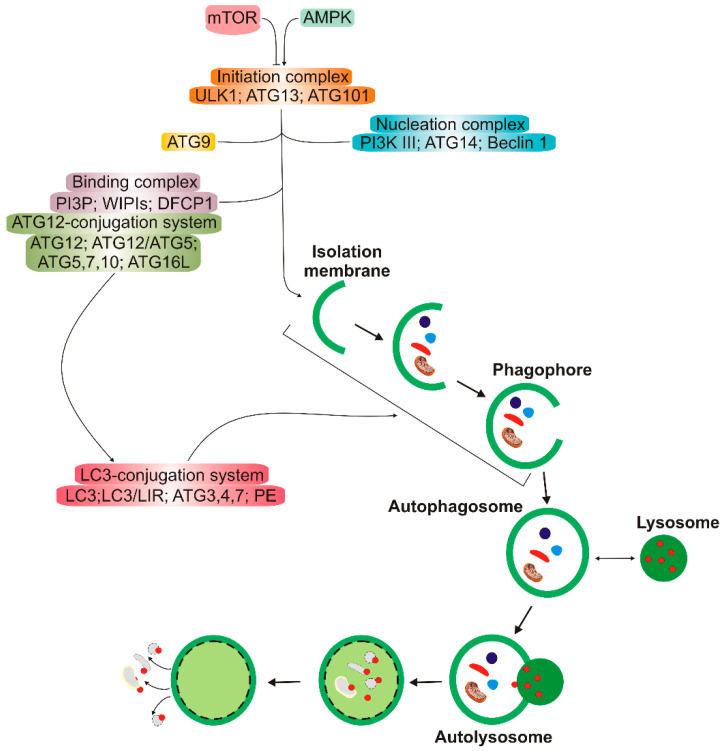
A simplified scheme of macro autophagy (further: autophagy). Mechanistic targets of rapamycin (mTOR) and AMP-activated kinase (AMPK) are the primary autophagy inhibitor and inducer, respectively. On autophagy initiation, the cytoplasmic material to be degraded (cargo) is gradually encapsulated by double membranes: phagophores to encapsulated vesicles (autophagosomes), which then fuse with lysosomes to form autolysosomes, where cargo is degraded and then released into cytoplasm to be used again (recycling). Autophagy is regulated by many proteins, the autophagy-related proteins (ATGs), and their complexes. The initiation process is led by the Unc − 51-like kinase 1 (ULK1) complex, the formation of the double membrane, provided by ATG − 9-containing vesicles, and autophagosome—by the class III PI3K (phosphatidylinositol 3-kinase) nucleation complex and the phosphatidylinositol 3-phosphate (PI3P)-binding complex, which includes ATG12, Beclin 1, and the microtubule-associated protein light chain 3/γ-aminobutyric acid receptor-associated proteins (LC3/GABARAPs), represented here only by LC3. ATG12 attaches to ATG5, which is then associated with ATG16L1. This complex stimulates the cleavage of LC3 by ATG4 to form LC3-I, which is then conjugated with phosphatidylethanolamine (PE) to form LC3-II. On the incorporation into autophagosomal membranes, LC3 interacts with cargo receptors, which have LC3-interacting motifs (LIRs).

**Figure 3 ijms-21-04994-f003:**
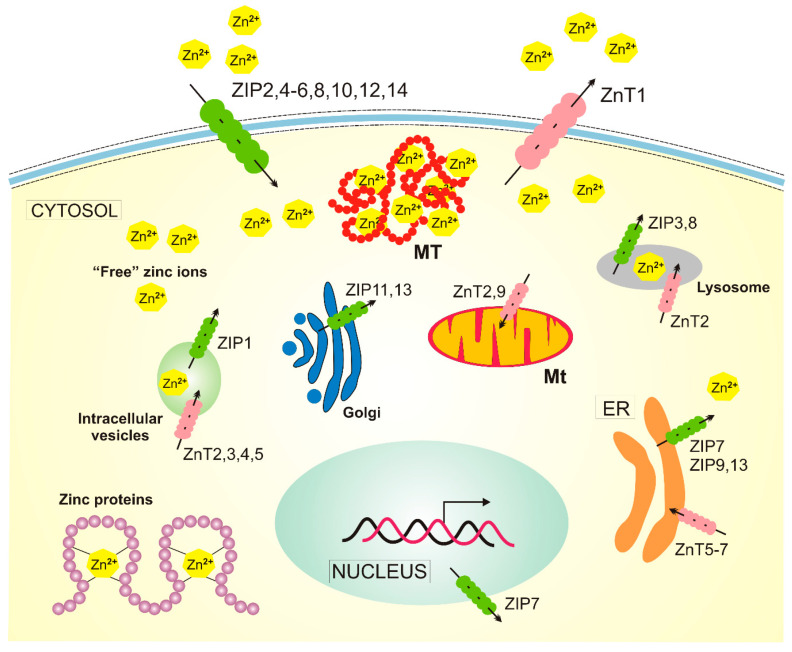
Cellular homeostasis of zinc in humans. There are at least 14 ZIP (Zn^2+^-regulated metal transporter, Iron-regulated metal transporter-like protein) proteins responsible for the transport of zinc ions into the cytosol and 10 zinc transporters, ZnTs (Zn^2+^ Transporter), acting in a reverse direction. Zinc is present in many human proteins, and metallothioneins (MTs) release zinc into the cytosol in oxidative conditions. The transport of zinc in/out of mitochondria (Mt) is not completely understood, and ZnT2 and ZnT9 presented here are likely not the only ones involved. Not all zinc transporters are presented in this figure, as many of them are expressed only in specific cells, such as ZnT8, which is specific to pancreatic cell islets. ER — endoplasmic reticulum.

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
