# Peer review of "Zinc and Autophagy in Age-Related Macular Degeneration"

_ijms, 2020, doi:10.3390/ijms21144994_

Round 1
Reviewer 1 Report
In this review, the authors are proposing a causative link between zinc, autophagy and age-related macular degeneration (AMD), a common eye disease causing blindness in many developed countries. As AMD severely affects patients' living quality, it is urgent to seek for therapeutics. Zinc as an essential trace element has been shown to potentially reduce progress of AMD, but the underlying mechanism(s) is unknown. The authors propose that the role of zinc in AMD is partially related to its function in autophagy. It is an interesting point of view and as a perspective it is interesting enough to inspire the audience to conduct relevant research in this direction. So this reviewer thinks this is a timely work and has been well written. Some statements about zinc biology need to be corrected, which are shown below.
Line 184, "most common transition metal". Suggest to replace "common" with "abundant".
Line 191, "zinc can be either bound to proteins or occur in a free state". Not correct. In cells, zinc can be bound with macromolecules (proteins, nucleic acids, membranes) and small molecules (GSH, ATP, amino acids, ...).
Figure 4, several issues. ZIP6 and ZIP12 are plasma membrane proteins. ZIP1 and ZIP3 are generally not considered as intracellular ZIPs. ZIP13 is primarily in Golgi, which is missed in the figure. ZnT2 and ZnT4 are thought to be intracellular. To show MT binds multiple zinc ions, it would be better to show 7, instead of 3, zinc ions. The full name of the ZIP is either Zrt-/Irt-like protein, or zinc-regulated transporter, iron-regulated transporter like protein. This reviewer prefers the former for the sake of clarity.
Line 215, "acts as a proton donor". This is improper to claim zinc is a proton donor because it does not have a proton. Instead, zinc facilitates bound water deprotonation by lowering water pKa.
Line 217-218, zinc indeed plays structural roles in many proteins, but it is not the reason why it is the cofactor in enzymes. In most of the cases, enzyme bound zinc plays catalytic role and is directly involved in catalysis.
Line 211-212, not all ZnT or ZIP is expressed in tissue specific way. Some of them are broadly expressed.
Line 224, "300 mg/g dry tissue", which means at least 3% of zinc in wet tissue. This value appears to be way too high. Please have double a check in literature.
Author Response
In this review, the authors are proposing a causative link between zinc, autophagy and age-related macular degeneration (AMD), a common eye disease causing blindness in many developed countries. As AMD severely affects patients' living quality, it is urgent to seek for therapeutics. Zinc as an essential trace element has been shown to potentially reduce progress of AMD, but the underlying mechanism(s) is unknown. The authors propose that the role of zinc in AMD is partially related to its function in autophagy. It is an interesting point of view and as a perspective it is interesting enough to inspire the audience to conduct relevant research in this direction. So this reviewer thinks this is a timely work and has been well written. Some statements about zinc biology need to be corrected, which are shown below.
Comment: Line 184, "most common transition metal". Suggest to replace "common" with "abundant".
Answer: We have done so.
Comment: Line 191, "zinc can be either bound to proteins or occur in a free state". Not correct. In cells, zinc can be bound with macromolecules (proteins, nucleic acids, membranes) and small molecules (GSH, ATP, amino acids, ...).
Answer: We have applied quotation marks to the word “free” and removed the word >rather< from the subordinate clause. Therefore, we have changed the sentence:
“In humans, as in other mammals, zinc can be either bound to proteins or occur in a free state, which represents rather zinc ions bound to non-protein ligands (reviewed in [28]).”
into:
“In humans, as in other mammals, zinc can be either bound to proteins or occur in a “free” state, which represents zinc ions bound to non-protein ligands (reviewed in [28]).”
We have also used quotation marks in the word “free” in the subsequent sentence.
Comment: Figure 4, several issues. ZIP6 and ZIP12 are plasma membrane proteins. ZIP1 and ZIP3 are generally not considered as intracellular ZIPs. ZIP13 is primarily in Golgi, which is missed in the figure. ZnT2 and ZnT4 are thought to be intracellular. To show MT binds multiple zinc ions, it would be better to show 7, instead of 3, zinc ions.
Answer: We have applied all these changes.
Comment: The full name of the ZIP is either Zrt-/Irt-like protein, or zinc-regulated transporter, iron-regulated transporter like protein. This reviewer prefers the former for the sake of clarity.
Answer: According to IJMS rules, the first use of an abbreviation must be accompanied by its definition in full. Therefore, the use of “Zrt-/Irt-like protein” would give the same as “zinc-regulated transporter, iron-regulated transporter like protein”
Comment: Line 215, "acts as a proton donor". This is improper to claim zinc is a proton donor because it does not have a proton. Instead, zinc facilitates bound water deprotonation by lowering water pKa.
Answer: We have changed the sentence:
“Unlike other transition metals, zinc lacks biological redox activity and acts as a proton donor that makes it as the good enzymatic cofactor [30].”
into:
“Unlike other transition metals, zinc lacks biological redox activity and facilitates bound water deprotonation by lowering water pKa, which make it a good enzymatic cofactor [30].”
Comment: Line 217-218, zinc indeed plays structural roles in many proteins, but it is not the reason why it is the cofactor in enzymes. In most of the cases, enzyme bound zinc plays catalytic role and is directly involved in catalysis.
Answer: We have changed the sentence:
“This may be a consequence of the fact that zinc is found as a structural component in thousands human proteins constituting about 10% of the human proteome [32].”
into:
“Moreover, enzyme bound zinc plays catalytic role and is directly involved in catalysis and is found as a structural component in thousands human proteins constituting about 10% of the human proteome [32].”
Comment: Line 211-212, not all ZnT or ZIP is expressed in tissue specific way. Some of them are broadly expressed.
Answer: Surely, yes. However, as we do not know about a universal, i.e. tissue-independent, zinc transporter, we have removed that sentence.
Comment: Line 224, "300 mg/g dry tissue", which means at least 3% of zinc in wet tissue. This value appears to be way too high. Please have double a check in literature.
Answer: Should write "300 µg/g dry tissue” – we have corrected this error.
Reviewer 2 Report
The authors focused on zinc as an important modulator of AMD development. I think reports on AMD focusing on zinc is relatively rare, therefore the present manuscript is very important. They are giving a very detailed review and I can learn a lot from this review. However, the manuscript is long overall and there are duplicated contents and phrases. I recommend to revise the manuscript to make it more compact.
I think the structure of the manuscript is not straightforward. The main theme of this manuscript is the role of zinc as a key substance of AMD development, but this manuscript contains two major subjects; one is AMD and autophagy/zinc and the other is general function of zinc in the body (5. Zinc and Autophagy). They described AMD and zinc at first, and described general function of zinc and described AMD and zinc again. For example, why don’ t you describe general function of zinc at first, then describe zinc and AMD?
The readers may expect “7. Conclusions and Perspectives” as a summery of this manuscript, but this section has a new information with new references. It is confusing.
I prefer the information of an adverse effect of zinc. Please add the description of it.
Line 20 “Zinc can stimulate autophagy, impaired in AMD. Therefore, it is rational to speculate that zinc may increase autophagy declined in aging and AMD and in this way slow down its progression.” This sentences may be “Zinc can stimulate autophagy that is impaired in AMD. Therefore, it is rational to speculate that zinc may increase autophagy which is declined in aging and AMD and slow down its progression in this way.” By the way, does “its” mean aging, AMD or both?
Line 57 “reason” might be “cause”.
Author Response
The authors focused on zinc as an important modulator of AMD development. I think reports on AMD focusing on zinc is relatively rare, therefore the present manuscript is very important. They are giving a very detailed review and I can learn a lot from this review.
Comment: However, the manuscript is long overall and there are duplicated contents and phrases. I recommend to revise the manuscript to make it more compact.
Answer: That this right, but some duplications have been deliberately introduced to “refresh” the content and enable to read some parts of the manuscript independently.
Comment: I think the structure of the manuscript is not straightforward. The main theme of this manuscript is the role of zinc as a key substance of AMD development, but this manuscript contains two major subjects; one is AMD and autophagy/zinc and the other is general function of zinc in the body (5. Zinc and Autophagy). They described AMD and zinc at first, and described general function of zinc and described AMD and zinc again. For example, why don’ t you describe general function of zinc at first, then describe zinc and AMD?
Answer: The main purpose of this manuscript to provide evidence that autophagy may contribute to beneficial effect of zinc in AMD. Therefore, AMD is the central subject of the manuscript. That is why, we choose the going order of sections in the manuscript
Comment: The readers may expect “7. Conclusions and Perspectives” as a summery of this manuscript, but this section has a new information with new references. It is confusing.
Answer: That is not a conclusions section, but also it sketches the perspectives and that is why it contains some new information.
Comment: I prefer the information of an adverse effect of zinc. Please add the description of it.
Answer: As we stated, the main subject of this manuscript was to provide evidence on the contribution of autophagy to beneficial effects of zinc in AMD, so we do not to overload this manuscript with data that are not directly linked with this subjects. Also, studies reported adverse effects of zinc in AMD are rare and questionable. We have added the following fragment to the “Discussion and perspectives”:
“However, these negative studies are rare and their research design is questionable and that is why they were not considered in the context of autophagy in AMD in this manuscript [53-56].”
Comment: Line 20 “Zinc can stimulate autophagy, impaired in AMD. Therefore, it is rational to speculate that zinc may increase autophagy declined in aging and AMD and in this way slow down its progression.” This sentences may be “Zinc can stimulate autophagy that is impaired in AMD. Therefore, it is rational to speculate that zinc may increase autophagy which is declined in aging and AMD and slow down its progression in this way.” By the way, does “its” mean aging, AMD or both?
Answer: That is right! These two sentences together have little sense. We have changed them into the following single sentence:
“As zinc can stimulate autophagy that is declined in AMD, it is rational to assume that it can slow down its progression.”
Comment: Line 57 “reason” might be “cause”.
Answer: We have done so.